# A Physically-Motivated Quantisation of the Electromagnetic Field on Curved Spacetimes

**DOI:** 10.3390/e21090844

**Published:** 2019-08-30

**Authors:** Ben Maybee, Daniel Hodgson, Almut Beige, Robert Purdy

**Affiliations:** 1Higgs Centre for Theoretical Physics, School of Physics and Astronomy, The University of Edinburgh, Edinburgh EH9 3JZ, UK; 2The School of Physics and Astronomy, University of Leeds, Leeds LS2 9JT, UK

**Keywords:** quantum electrodynamics, relativistic quantum information

## Abstract

Recently, Bennett et al. (Eur. J. Phys. **37**:014001, 2016) presented a physically-motivated and explicitly gauge-independent scheme for the quantisation of the electromagnetic field in flat Minkowski space. In this paper we generalise this field quantisation scheme to curved spacetimes. Working within the standard assumptions of quantum field theory and only postulating the physicality of the photon, we derive the Hamiltonian, $\hat{H}$, and the electric and magnetic field observables, $\hat{E}$ and $\hat{B}$, respectively, without having to invoke a specific gauge. As an example, we quantise the electromagnetic field in the spacetime of an accelerated Minkowski observer, Rindler space, and demonstrate consistency with other field quantisation schemes by reproducing the Unruh effect.

## 1. Introduction

For many theorists the question “what is a photon?” remains highly nontrivial [1]. It is in principle possible to uniquely define single photons in free space [2]; however, the various roles that photons play in light–matter interactions [3], the presence of boundary conditions in experimental scenarios [4,5] and our ability to arbitrarily shape single photons [6] all lead to a multitude of possible additional definitions. Yet this does not stop us from utilising single photons for tasks in quantum information processing, especially for quantum cryptography, quantum computing, and quantum metrology [7]. In recent decades, it has become possible to produce single photons on demand [8], to transmit them over 100 kilometres through Earth’s atmosphere [9] and to detect them with very high efficiencies [10]. Moreover, single photons have been an essential ingredient in experiments probing the foundations of quantum physics, such as entanglement and locality [11,12].

Recently, relativistic quantum information has received a lot of attention in the literature. Pioneering experiments verify the possibility of quantum communication channels between Earth’s surface and space [13] and have transmitted photons between the Earth and low-orbit satellites [14], while quantum information protocols are beginning to extend their scope towards the relativistic arena [15,16,17,18,19,20,21]. The effects of gravity on satellite-based quantum communication schemes, entanglement experiments and quantum teleportation have already been shown to produce potentially observable effects [22,23,24,25]. Noninertial motion strongly affects quantum information protocols and quantum optics set-ups [26,27,28,29,30], with the mere propagation and detection of photons in such frames being highly nontrivial [31,32,33,34].

Motivated by these recent developments, this paper generalises a physically-motivated quantisation scheme of the electromagnetic field in flat Minkowski space [35] to curved space times. Our approach aims to obtain the basic tools for analysing and designing relativistic quantum information experiments in a more direct way than alternative derivations, and without having to invoke a specific gauge. Working within the standard assumptions of quantum field theory and only postulating the physicality of the photon, we derive the Hamiltonian, $\hat{H}$, and the observables, $\hat{E}$ and $\hat{B}$, of the electromagnetic field. Retaining gauge-independence is important when modelling the interaction of the electromagnetic field with another quantum system, like an atom. In this case, different choices of gauge correspond to different subsystem decompositions, thereby affecting our notion of what is ‘atom’ and what is ‘field’ [36,37]. Composite quantum systems can be decomposed into subsystems in many different ways. Choosing an unphysical decomposition can result in the prediction of spurious effects when analysing the dynamics of one subsystem while tracing out the degrees of freedom of the other [38]. Hence it is important to first formulate quantum electrodynamics in an entirely arbitrary gauge, as this allows us to subsequently fix the gauge when needed. This work does not seek to quantise the gravitational field. Instead, we follow the standard approach of quantum field theory in curved spacetime. This is a first approximation to understanding gravitational effects on quantum fields [39,40], which neglects the back-reaction of those fields on the spacetime geometry, treating the spacetime as a fixed background.

The direct canonical quantisation of the electromagnetic field in terms of the (real) gauge independent electric and magnetic fields, E and B, is not possible, since these do not offer a complete set of canonical variables [41,42,43,44,45]. As an alternative, Bennett et al. [35] suggested to use the physicality of the photon as the starting point when quantising the electromagnetic field. Assuming that the electromagnetic field is made up of photons and identifying their relevant degrees of freedom, like frequencies and polarisations, results in a harmonic oscillator Hamiltonian $\hat{H}$ for the electromagnetic field. Using this Hamiltonian and demanding consistency of the dynamics of expectation values with classical electrodynamics, especially with Maxwell’s equations, is sufficient to then obtain expressions for $\hat{E}$ and $\hat{B}$ without having to invoke vector potentials and without having to choose a specific gauge. Generalising the work by the authors of [35] from flat Minkowski space to curved space times, we obtain field observables which could be used, for example, to model photonics experiments in curved spacetimes in a similar fashion to how quantum optics typically models experiments in Minkowski space [5,36,46].

Additional problems with our understanding of photons (indeed all particles) arise when we consider quantum fields in gravitationally bound systems [7]. General relativity can be viewed as describing gravitation as the consequence of interactions between matter and the curvature of a Lorentzian (mixed signature) spacetime with metric $g_{\mu \nu }$ [47,48]. Locally, however, any spacetime appears flat, by which we mean
(1)$$g_{\mu \nu }\left(p\right)≅\eta_{\mu \nu }\equiv \mathrm{diag}\left(+1,-1,-1,-1\right)\;,$$
the familiar special relativistic invariant line-element of Minkowski space. For the Earth’s surface, where gravity is (nearly) uniform, this limit can be taken everywhere, and spacetime curvature can be neglected. Spacetimes in relativity have no preferred coordinate frame, so physical laws must satisfy the principle of covariance and be coordinate independent and invariant under coordinate transformations [49]. Indeed, it has been demonstrated that, while the form of the Hamiltonian may change under general coordinate transformations, physically measurable predictions do not [50].

Quantum field theory in curved spacetime is the standard approach used to study the behaviour of quantum fields in this setting. As aforementioned, this is a first approximation to quantum gravity, in which the gravitational field is treated classically and back-reactions on the spacetime geometry are neglected [39,40]. Intuitively this is what is meant by a static spacetime, where the time derivative of the metric is zero. This approximation holds on typical astrophysical length and energy scales and is thus well-suited for dealing with most physical situations [51]. How to generalise field quantisation to curved spaces is very well established, and the theory has produced several major discoveries, like the prediction that the particle states seen by a given observer depend on the geometry of their spacetime [52,53,54]. For example, the vacuum state of one observer does not necessarily coincide with the vacuum state of an observer in an alternative reference frame. This surprising result even arises in flat Minkowski space, where the Fulling–Davies–Unruh effect predicts that an observer with constant acceleration sees the Minkowski vacuum as a thermal state with temperature proportional to their acceleration [55,56,57,58,59].

To make quantum field theory in curved spacetimes more accessible to quantum opticians, and to obtain more insight into the aforementioned effects and their experimental ramifications, this paper considers static, 4-dimensional Lorentzian spacetimes. Our starting point for the derivation of the field observables $\hat{H}$, $\hat{E}$ and $\hat{B}$ is the assumption that the detectors belonging to a moving observer see photons. These are the energy quanta of the electromagnetic field in curved space times. To demonstrate the consistency of our approach with other field quantisation schemes, we consider the explicit case of an accelerated Minkowski observer, who is said to reside in a Rindler spacetime [60,61,62,63,64], and reproduce Unruh’s predictions [55,56,57,58,59].

This paper is divided into five sections. In Section 2, we provide a summary of the gauge-independent quantisation scheme by Bennett et al. [35] which applies in the case of flat spacetime. In Section 3, we discuss what modifications must be made to classical electrodynamics when moving to the more general setting of a stationary curved spacetime. We then show that similar modifications allow for the gauge-independent quantisation scheme of Section 2 to be applied in this more general setting. In Section 4, we apply our results to the specific case of a uniformly accelerating reference frame and have a closer look at the Unruh effect. Finally, we draw our conclusions in Section 5. For simplicity, we work in natural units ℏ=c=1 throughout.

## 2. Gauge-Independent Quantisation of the Electromagnetic Field

In this section, we review the gauge dependence inherent in the electromagnetic field and contrast standard, more mathematically-motivated quantisation procedures with the gauge-independent method of Bennett et al. [35].

### 2.1. Classical Electrodynamics

Under coordinate transformations, the electric and magnetic fields transform as the components of an antisymmetric 2-form, the field strength tensor
(2)$$F_{\mu \nu }=\left(\begin{array}{cccc} 0 & E^{1} & E^{2} & E^{3}\\ -E^{1} & 0 & -B^{3} & B^{2}\\ -E^{2} & B^{3} & 0 & -B^{1}\\ -E^{3} & -B^{2} & B^{1} & 0 \end{array}\right).$$
The field strength is defined in terms of the 4-vector potential by
(3)$$F_{\mu \nu }=\partial_{\mu }A_{\nu }-\partial_{\nu }A_{\mu }\;.$$
We can obtain the equations of motion by applying the Euler–Lagrange equations to the Lagrangian density
(4)$$L=-\frac{1}{4}F_{\mu \nu }F^{\mu \nu }=\frac{1}{2}\left(E^{2}-B^{2}\right),$$
which gives the Maxwell equation
(5)$$\partial_{\mu }F^{\mu \nu }=0\;.$$
The field strength tensor also satisfies the Bianchi identity,
(6)$$\partial_{[\sigma }F_{\mu \nu ]}\equiv \frac{1}{3}\left(\partial_{\sigma }F_{\mu \nu }+\partial_{\mu }F_{\nu \sigma }+\partial_{\nu }F_{\sigma \mu }\right)=0\;,$$
and together, Equations (Equation 5) and (Equation 6) can be used to obtain the standard Maxwell equations expressed in terms of the magnetic and electric field strengths, E and B, respectively,
(7)$$\begin{array}{cc} \mathrm{div}\left(E\right)=0\;, & \;\mathrm{curl}\left(B\right)=\dot{E}\;,\\ \mathrm{div}\left(B\right)=0\;, & \;\mathrm{curl}\left(E\right)=-\dot{B}\;. \end{array}$$
The solutions to these equations are transverse plane waves with orthogonal electric and magnetic field components with two distinct, physical polarisations propagating through Minkowski space, M, at a speed c=1.

### 2.2. Gauge Dependence in Electromagnetic Field Quantisation

The most commonly used methods for quantising fields are the traditional canonical and modern path-integral approaches. When applied to electromagnetism, these have to be modified due to the gauge freedom of the theory. For example, in the canonical approach, standard commutation relations cannot be satisfied. One can get around this by either breaking Lorentz invariance in intermediate steps of calculations, or by considering excess degrees of freedom with negative norms that do not contribute physically [37]. Standard path integral quantisation fails for electromagnetism because the resultant propagator is divergent. The Fadeev–Popov procedure rectifies this by implementing a gauge-fixing condition [65]. This method also gives additional terms from nonphysical contributions in the form of Fadeev–Popov ghosts. Such terms can be ignored for free fields in Minkowski space as they only appear in loop diagrams, but in curved spacetimes this is not the case [51]. While physical quantities remain gauge-invariant under both approaches to quantisation, nondirectly observable quantities can become gauge-dependent.

This can result in conceptual problems when modelling composite quantum systems, like the ones that are of interest to those working in relativistic quantum information, quantum optics and condensed matter. Suppose *H* denotes the total Hamiltonian of a composite quantum system. Then one can show that any Hamiltonian $H^{\prime }$ of the form
(8)$$\begin{array}{ccc} H^{\prime } & = & U^{†}\;H\;U\;, \end{array}$$
where *U* denotes a unitary operator, has the same energy eigenvalues as *H*. Both Hamiltonians *H* and $H^{\prime }$ are unitarily equivalent and can be used interchangeably. However, the dynamics of subsystem observables *O* can depend on the concrete choice of *U*, since $O^{\prime }=U^{†}\;O\;U$ and *O* are in general not the same. For example, atom–field interactions depend on the gauge-dependent vector potential A for most subsystem decompositions [36,37]. Hence it is important here to formulate quantum electrodynamics in an entirely arbitrary gauge and to maintain ambiguity as long as possible, thereby retaining the ability to later choose a gauge which does not result in the prediction of spurious effects [38].

### 2.3. Physically-Motivated Gauge-Independent Method

In contrast to this, the electromagnetic field quantisation scheme presented in the work by the authors of [35] relies upon two primary experimentally derived assumptions. Firstly, the electric and the magnetic field expectation values follow Maxwell’s equations, and, secondly, the field is composed of photons of energy $\hbar \omega_{k}$, or $\omega_{k}$ in natural units. Whereas, in standard canonical quantisation, the electromagnetic field’s photon construction is a derived result, for the method of [35] it is an initial premise. This is physically acceptable, since photons are experimentally detectable entities [7,10]. The motivation for the scheme [35] comes from the observation that one observes discrete clicks when measuring a very weak electromagnetic field. An experimental definition of photons is that these are electromagnetic field excitations with the property that their integer numbers can be individually detected, given a perfect detector [10].

Hence the Fock space for this gauge-independent approach is spanned by states of the form
(9)$$\underset{\lambda =1,2}{⨂}\overset{\infty }{\underset{k^{1}=-\infty }{⨂}}\overset{\infty }{\underset{k^{2}=-\infty }{⨂}}\overset{\infty }{\underset{k^{3}=-\infty }{⨂}}|n_{k\lambda }\rangle \;,$$
where $n_{k\lambda }$ is the number of excitations of a mode with wave-vector k and physical, transverse polarisation state λ. Since it is an experimental observation that photons of frequency $\omega_{k}=\left|k\right|$ have energy $\omega_{k}$ in natural units, the Hamiltonian $\hat{H}$ for such a Fock space must satisfy
(10)$$\hat{H}\;|n_{k\lambda }\rangle =\left(\omega_{k}\;n_{k\lambda }+H_{0}\right)|n_{k\lambda }\rangle \;,$$
where $H_{0}$ is the vacuum or zero-point energy and $n_{k\lambda }$ is an integer [35]. An infinite set of evenly spaced energy levels, as is present here, has been proven to be unique to the simple harmonic oscillator [66]. Hence this Hamiltonian must take the form [5]
(11)$$\hat{H}=\sum_{\lambda =1,2}\int d^{3}k\left(\omega_{k}\;{\hat{a}}_{k\lambda }^{†}{\hat{a}}_{k\lambda }+H_{0}\right)\;,$$
where the ${\hat{a}}_{k\lambda },{\hat{a}}_{k\lambda }^{†}$ are a set of independent ladder operators for each (k,λ) mode, obeying the canonical commutation relations
(12)$$\begin{array}{c} \left[{\hat{a}}_{k\lambda },{\hat{a}}_{k^{\prime }\lambda^{\prime }}\right]=0\;,\;\left[{\hat{a}}_{k\lambda }^{†},{\hat{a}}_{k^{\prime }\lambda^{\prime }}^{†}\right]=0\;,\;\left[{\hat{a}}_{k\lambda },{\hat{a}}_{k^{\prime }\lambda^{\prime }}^{†}\right]=\delta_{\lambda \lambda^{\prime }}\delta^{3}\left(k-k^{\prime }\right)\;. \end{array}$$


Since the classical energy density is quadratic in the electric and magnetic fields, while the above Hamiltonian is quadratic in the ladder operators, the field operators must be linear superpositions of creation and annihilation operators [35]. By further demanding that the fields’ expectation values satisfy Maxwell’s equations, consistency with the Heisenberg equation of motion,
(13)$$\frac{\partial }{\partial t}\hat{O}=-i\left[\hat{O},\hat{H}\right]\;,$$
allows the coefficients of these superpositions to be deduced, and the (Heisenberg) field operators can be shown to be of the form [35]
(14)$$\begin{array}{ccc} \hat{E}\left(x,t\right) & = & i\sum_{\lambda =1,2}\int d^{3}k\sqrt{\frac{\omega_{k}}{16\pi^{3}}}\left[e^{i(k\cdot x-\omega_{k}t)}{\hat{a}}_{k\lambda }+H.c.\right]{\hat{e}}_{\lambda }\;,\\ \hat{B}\left(x,t\right) & = & -i\sum_{\lambda =1,2}\int d^{3}k\sqrt{\frac{\omega_{k}}{16\pi^{3}}}\left[e^{i(k\cdot x-\omega_{k}t)}{\hat{a}}_{k\lambda }+H.c.\right]\left(\hat{k}\times {\hat{e}}_{\lambda }\right)\;, \end{array}$$
where $\hat{e}{}_{\lambda }$ is a unit polarisation vector orthogonal to the direction of propagation, with ${\hat{e}}_{1}\cdot {\hat{e}}_{2}={\hat{e}}_{1}\cdot k={\hat{e}}_{2}\cdot k=0$. This is also consistent with the Hamiltonian being a direct operator-valued promotion of its classical form
(15)$$\hat{H}\left(t\right)=\frac{1}{2}\int d^{3}x\;\left[\hat{E}{}^{2}\left(x,t\right)+\hat{B}{}^{2}\left(x,t\right)\right]\;.$$
Comparing Equations (Equation 11) and (Equation 15) allows us to determine the zero point energy $H_{0}$ in Minkowski space, which coincides with the energy expectation value of the vacuum state |0〉 of the electromagnetic field. In quantum optics, Equations (Equation 11) and (Equation 14) often serve as the starting point for further investigations [5,36,46].

Note that a quantisation scheme in a similar spirit to the work by the authors of [35] can be found in the work by the authors of [67], which also uses the Maxwell and Heisenberg equations to directly quantise the physical field operators. The attraction of such a scheme is in the lack of reliance on the gauge-dependent electromagnetic potentials, instead directly quantising the gauge-invariant electric and magnetic fields, the benefits of which for quantum optics were discussed in the preceding section.

## 3. Gauge-Independent Quantisation of the Electromagnetic Field in Curved Spacetimes

Many aspects of the quantisation method of Bennett et al. [35] are explicitly noncovariant, and hence unsuitable for general curved spacetimes. Here we lift the scheme onto static spacetimes, maintaining the original global structure and approach.

### 3.1. Classical Electrodynamics in Curved Space

To begin, consider electromagnetism on stationary spacetimes in general relativity, which are differentiable manifolds with a metric structure $g_{\mu \nu }$. By stationary we mean $\partial_{0}g_{\mu \nu }=0$. For any theory, the standard approach is to follow the minimal-coupling procedure [48,51],
(16)$$\begin{array}{c} \eta_{\mu \nu }\to g_{\mu \nu }\;,\\ \partial_{\mu }\to \nabla_{\mu }\;,\\ \int d^{4}x\to \int d^{4}x\;\sqrt{\left|g\right|}\;, \end{array}$$
where $g=\det(g_{\mu \nu })$ and $\nabla_{\mu }$ is the covariant derivative associated with the metric (Levi-Civita) connection. Since electric and magnetic fields can be expressed in a covariant form through the field strength tensor, it is simple to generalise to curved space by just applying this procedure. Firstly, the derivatives of the four-vector potential generalise to
(17)$$\begin{array}{cc} \nabla_{\nu }A_{\mu } & =\partial_{\nu }A_{\mu }-\Gamma_{\mu \nu }^{\rho }A_{\rho }\;,\\ \nabla_{\nu }A^{\mu } & =\partial_{\nu }A^{\mu }+\Gamma_{\rho \nu }^{\mu }A^{\rho }\;, \end{array}$$
where $\Gamma_{\nu \rho }^{\mu }$ are the standard symmetric Christoffel symbols. The field strength tensor and the Bianchi identity remain unchanged by these derivatives, as their explicit antisymmetry cancels all the Christoffel symbols. Thus Equations (Equation 3) and (Equation 6) still hold in curved spacetimes. The only modification we need to make is to the (free-space) inhomogeneous Maxwell equation. Applying the minimal-coupling procedure to Equation (Equation 5) gives
(18)$$\nabla_{\mu }F^{\mu \nu }=0\;,$$
which on stationary spacetimes can be written as [61]
(19)$$\nabla_{\mu }F^{\mu \nu }=\frac{1}{\sqrt{\left|g\right|}}\partial_{\mu }\left(\sqrt{\left|g\right|}F^{\mu \nu }\right)=0\;,$$
as may be obtained from a Lagrangian density $L=-\frac{1}{4}\sqrt{\left|g\right|}F_{\mu \nu }F^{\mu \nu }$. To obtain the modified Maxwell equations for the electric and magnetic field strengths, one may now simply extract the relevant terms from the covariant form given above, working in a particular coordinate system [60]. For the resulting wave equations, as with any wave equation on a curved spacetime, obtaining a general solution is a highly nontrivial task [49]. However, on simple spacetimes such as we will consider later, it is possible to obtain analytic solutions.

### 3.2. Particles in Curved Spacetimes

To quantise the electromagnetic field in the manner of [35] our starting point must be to write down an appropriate Fock space for experimentally observable photon states. On curved spacetimes this is complicated by the lack of a consistent frame-independent basis for such a space. To see why, consider that to introduce particle states in quantum field theory, we must first write the solutions to a momentum–space wave equation as a superposition of orthonormal field modes, which are split into positive and negative frequency modes ($f_{i},f_{i}^{*}$). In order for us to do this, the spacetime must have a timelike symmetry. Symmetries of spacetimes are generated by Killing vectors, *V*, which satisfy
(20)$$\nabla_{\mu }V_{\nu }+\nabla_{\nu }V_{\mu }=0\;.$$
If $V_{\mu }$ is, in addition, timelike at asymptotic infinity then it defines a timelike Killing vector $K_{\mu }$. The presence of such a vector defines a stationary spacetime, in which there always exists a coordinate frame such that $\partial_{t}g_{\mu \nu }=0$, where $x^{0}=t$ in this coordinate set is the Killing time. If, in addition, $K^{\mu }$ is always orthogonal to a family of spacelike hypersurfaces then the spacetime is said to be static, and in addition we have $g_{ti}=0$. Conceptually, the spacetime background is fixed but fields can propagate and interact. Particle states can only be canonically introduced with frequency splitting. Hence, to define particles in a curved spacetime there must be a timelike Killing vector [53].

Canonical field quantisation morphs the field into an operator acting on a Fock space of particle states, promoting the coefficients of the positive frequency modes to annihilation operators and those of negative frequency modes to creation operators [33,40]. General field states are therefore critically dependent on the frequency splitting of the modes, which itself depends on the background geometry of the spacetime [52]. In general, we define positive and negative frequency modes $f_{\omega_{k}}$ of frequency $\omega_{k}$ with respect to the timelike Killing vector $K^{\mu }$, by using the definition
(21)$${£}_{K}f_{\omega_{k}}=\left\{\begin{array}{ccc} -i\omega_{k}f_{\omega_{k}} & \; & \mathrm{positive}\;\mathrm{frequency}\\ i\omega_{k}f_{\omega_{k}} & \; & \mathrm{negative}\;\mathrm{frequency} \end{array}\right.\;,$$
where ${£}_{K}$ is the coordinate-invariant Lie derivative along $K^{\mu }$, which, in this case, is given by $K^{\mu }\partial_{\mu }$. However, a particle detector reacts to states of positive frequency with respect to its own proper time τ, not the killing time [55]. For a timelike observer with worldline $x^{\mu }$ on a (not necessarily stationary) spacetime, the proper time is defined by the metric $g_{\mu \nu }$ infinitesimally as
(22)$$d\tau =\sqrt{g_{\mu \nu }dx^{\mu }dx^{\nu }}\;.$$
A given detector with proper time τ has positive frequency modes $g_{\omega_{k}}$ satisfying
(23)$$\frac{dx^{\mu }}{d\tau }\nabla_{\mu }g_{\omega_{k}}=-i\omega_{k}g_{\omega_{k}}\;,$$
and they will, generally, only cover part of the spacetime. To consistently approach quantisation we need these detector modes to relate to the set $f_{\omega_{k}}$ defined with respect to the timelike Killing vector. Fortunately, the set of modes $f_{\omega_{k}}$ forms a natural basis for the detector’s Fock space if the proper time τ is proportional to the Killing time *t*. This occurs if the future-directed timelike Killing vector is tangent to the detector’s trajectory [48,55].

Even with a timelike Killing vector and its associated symmetry, solving a given wave equation and hence obtaining mode solutions can still be highly nontrivial [49]. Considering a static spacetime greatly simplifies this as the d’Alembertian operator, $□=\nabla_{\mu }\nabla^{\mu }$, that appears in the general wave equation can be separated into pure spatial and temporal derivatives, allowing us to easily write separable mode solutions [48,52]
(24)$$f_{\omega_{k}}\left(x\right)=e^{-i\omega_{k}t}\Sigma_{\omega_{k}}\left(x\right)\;.$$
These modes are then positive frequency in the above sense, and conjugate modes $f_{\omega_{k}}^{*}$ are negative frequency. The set $\left(f_{\omega_{k}},f_{\omega_{k}}^{*}\right)$ then forms a complete basis of solutions for the wave equation and provides a suitable basis for particle detectors.

However, when two distinct inertial particle detectors follow different geodesic paths in the spacetime, each will have its own unique proper time, determined by its motion and the local geometry. But this proper time is what we have used in Equation (Equation 23) to define the basis modes associated with a given particle Fock space associated with a particle detector. Thus, the detectors will define the particle states they observe in different manners, and will not agree on a natural set of basis modes [52,53,68]. This has no counterpart in inertial Minkowski space, where there is a global Poincaré symmetry, but will be unavoidable in our scheme.

### 3.3. Covariant and Gauge-Independent Electromagnetic Field Quantisation Scheme

Accommodating for the above considerations allows the physically motivated scheme [35] to be covariantly generalised to static curved spacetimes.

#### 3.3.1. Hilbert Space

Since for static spacetimes there exists a global timelike Killing vector we can define positive and negative frequency modes and thus introduce a well-defined particle Fock space. Again we assume the existence of photons on the considered spacetime. As travelling waves on the spacetime, these photons are again characterised by their physical, transverse polarisation λ and wave-vector k [69]. Taking these as labels for general states yields again the states in Equation (Equation 9) as the basis states of the quantised field. Physical energy eigenstates have integer values of $n_{k\lambda }$ and are associated with energy $\omega_{k}$. Thus the field Hamiltonian must again satisfy Equation (Equation 10), allowing it to be written in terms of independent ladder operators [66]. In the following, we denote these by $b_{k\lambda }$ and assume that they satisfy the equal time canonical commutation relations
(25)$$\begin{array}{c} \left[{\hat{b}}_{k\lambda },{\hat{b}}_{k^{\prime }\lambda^{\prime }}\right]=0\;,\;\left[{\hat{b}}_{k\lambda }^{†},{\hat{b}}_{k^{\prime }\lambda^{\prime }}^{†}\right]=0\;,\;\left[{\hat{b}}_{k\lambda },{\hat{b}}_{k^{\prime }\lambda^{\prime }}^{†}\right]=\delta_{\lambda \lambda^{\prime }}\delta^{3}\left(k-k^{\prime }\right)\;. \end{array}$$
Importantly, the $b_{k\lambda }$ generate a distinct Fock space from that of the ladder operators utilised in the Minkowskian case.

#### 3.3.2. Hamiltonian

To write down the full field or classical Hamiltonian requires some care, as a Hamiltonian is a component of the energy–momentum tensor
(26)$$T_{\mu \nu }=-\frac{2}{\sqrt{\left|g\right|}}\frac{\delta S_{\mathrm{matter}}}{\delta g^{\mu \nu }}\;,$$
where $S_{\mathrm{matter}}$ is the action determining the matter content on the spacetime. As a component of a tensor, the Hamiltonian itself is not invariant under general coordinate transformations. On stationary spacetimes a conserved energy equal to the Hamiltonian can be introduced through the timelike Killing current
(27)$$J^{\mu }=K_{\nu }T^{\mu \nu }\;,$$
which satisfies the continuity equation $\nabla_{\mu }J^{\mu }=0$. Stokes’s theorem can then be used to integrate over a spacelike hypersurface Σ in three dimensions, giving
(28)$$H=\int_{\Sigma }d^{3}x\;\sqrt{\left|\gamma \right|}n_{\mu }J^{\mu }\;,$$
where $\gamma =\det(\gamma_{ij})$ with $\gamma_{ij}$ being the induced metric on Σ and $n^{\mu }$ being the timelike unit normal vector to Σ. On stationary spacetimes the result of this integral is the same for all hypersurfaces Σ [48,70]. For the electromagnetic field, the variation in Equation (Equation 26) yields
(29)$$T_{\mu \nu }=F_{\mu \rho }F^{\rho }{}_{\nu }+\frac{1}{4}g_{\mu \nu }F_{\rho \sigma }F^{\rho \sigma }\;,$$
from which we can obtain a covariant form of the classical electromagnetic Hamiltonian.

Note that, since in Equation (Equation 28) Σ is a spacelike hypersurface, $n^{\mu }$ must be timelike. Thus, there exists a frame in which $n_{\mu }J^{\mu }=n_{0}J^{0}$, and as this is a scalar this is valid in any frame. We also have that $J^{\mu }=T_{0}^{\mu }$, so we seek $T_{0}^{0}$. On a static spacetime $T_{0}^{0}=g^{00}T_{00}$, so
(30)$$T_{0}^{0}=\frac{1}{2}\left(E^{2}+B^{2}\right)\;,$$
where in the intermediate step we have used the Minkowski field strength tensor, as the quantities are scalars. Hence we obtain the electromagnetic field Hamiltonian
(31)$$H=\int_{\Sigma }d^{3}x\;\frac{1}{2}\left(E^{2}+B^{2}\right)\sqrt{\left|\gamma \right|}n_{0}K^{0}\;.$$
This result is consistent with the literature [47], and reduces to the familiar expression in Equation (Equation 15) in Minkowski space.

For the covariant analogue of the quantum field Hamiltonian, we note that the field Hamiltonian used in the Minkowskian gauge-independent scheme, given in Equation (Equation 11), has a similar functional form to the Hamiltonian for a quantised scalar field; they are identical up to labelling and choice of integration measure. It has been established by Friis et al. [16] that the propagation of transverse electromagnetic field modes can be well approximated by such an uncharged field, and this technique has been used to determine the effects of spacetime curvature on satellite-based quantum communications and to make metrology predictions [23,26]. In the following, we use this approximation to justify the form of the electromagnetic field Hamiltonian from that of a real scalar field with the equation of motion $(□+m^{2})\phi =0$. The Hamiltonian density on a static manifold with Killing time *t* is
(32)$$\begin{array}{c} H=\frac{\;\sqrt{\left|g\right|}}{2}\left(\partial^{t}\phi \partial_{t}\phi -\partial^{i}\phi \partial_{i}\phi +\frac{1}{2}m^{2}\phi^{2}+\frac{1}{2}\xi R\phi \right). \end{array}$$
The final term pertains to the coupling between the spacetime background and the field. Given we just seek to study photons propagating on some curved background and are ignoring their back-reaction on the geometry, we can choose ξ=0. This is known as the minimal coupling approximation.

Since on static spacetimes the d’Alembertian permits separable solutions, we can write $\phi =\psi_{\omega_{k}}\left(x\right)e^{\pm iE_{\omega_{k}}t}$ [48,52]. Here $\psi_{\omega_{k}},E_{\omega_{k}}$ are the eigenstates of the Klein–Gordon operator $\left(□+m^{2}\right)$. Upon quantisation, the field operator for a real scalar field can now be written as a linear superposition of these modes with ladder operators $b_{\omega_{k}},b_{\omega_{k}}^{†}$ defining the Fock space. However, we must also account for the nonuniqueness of particle states in curved spacetimes. One set of Fock space operators is often not able to cover an entire spacetime, so we will include a sum over distinct sets of operators, $b_{\omega_{k}}^{\left(i\right)},b_{\omega_{k}}^{(i)†}$. Following Fulling [52], we introduce a measure, $\mu (\omega_{k})$, such that if the eigenstates form a complete basis for the Hilbert space of states, allowing a general function to be written as $F\left(x\right)=\int d\mu \left(\omega_{k}\right)\left(f\left(\omega_{k}\right)\psi_{\omega_{k}}\left(x\right)\right)$, the inner product on the Hilbert space becomes
(33)$$\langle F_{1},F_{2}\rangle =\int d^{3}x\sqrt{\left|g\right|}g^{tt}F_{1}^{*}F_{2}=\int d\mu \left(\omega_{k}\right)f_{1}^{*}f_{2}\;.$$
With this measure, the Hamiltonian field operator for a minimally-coupled scalar field on any static spacetime can be written as [52]
(34)$$\hat{H}=\int d\mu \left(\omega_{k}\right)\sum_{i}E_{\omega_{k}}^{\left(i\right)}\left[{\hat{b}}_{\omega_{k}}^{(i)†}{\hat{b}}_{\omega_{k}}^{\left(i\right)}+\frac{1}{2}\delta \left(0\right)\right]\;.$$
Thus using the approximation of Friis et al. [16], we obtain the same functional form for the free electromagnetic quantised Hamiltonian on any static spacetime. To incorporate the direction of propagation, we can instead label modes in the above expressions by their wave-vector satisfying $\left|k\right|=\omega_{k}$. Then the integration measure $\mu (\omega_{k})$ can be taken as $d\mu \left(\omega_{k}\right)=d^{3}k$. This applies since
(35)$$F\left(x\right)=\int d^{3}k\left(f\left(k\right)\psi_{k}\left(x\right)\right)$$
and the inner product of two such functions is
(36)$$\begin{array}{ccc} \langle F_{k},F_{k^{\prime }}\rangle & = & \int d^{3}k\int d^{3}k^{\prime }\int d^{3}x\;\sqrt{\left|g\right|}g^{tt}f_{k}^{*}f_{k^{\prime }}\psi_{k}^{*}\psi_{k^{\prime }}\\ & = & \int d^{3}k\int d^{3}k^{\prime }\;f_{k}^{*}f_{k^{\prime }}\langle \psi_{k}^{*},\psi_{k^{\prime }}\rangle \\ & = & \int d^{3}k\;f_{k}^{*}f_{k}\;. \end{array}$$
To obtain the third line we have used that $\psi_{k}$ and $\psi_{k^{\prime }}$ are eigenstates of a self-adjoint operator [52]. Physical photon modes will also be indexed by their transverse polarisation, so we also introduce an additional mode label for the polarisation λ. Thus, in all, for a minimally-coupled electromagnetic field on a static Lorentzian manifold the quantised field Hamiltonian for the Fock space defined in Equation (Equation 9) can be taken as
(37)$$\hat{H}=\sum_{\lambda =1,2}\int d^{3}k\left(\sum_{i}\omega_{k}^{\left(i\right)}\;{\hat{b}}_{k\lambda }^{(i)†}{\hat{b}}_{k\lambda }^{\left(i\right)}+H_{0}\right)\;.$$
Other than the sum over distinct sectors, this result is no different from its Minkowskian counterpart; this has only been possible with careful considerations of the static curved background.

#### 3.3.3. Electromagnetic Field Observables

The classical Hamiltonian remains quadratic in the electric and magnetic fields, and the quantised field Hamiltonian is still quadratic in the ladder operators. As is demonstrated above, this will continue to be the case for any static spacetime, as it was in the Minkowskian case of Section 2.3. In nonstatic spacetimes, the lack of a conserved local energy introduces ambiguity into our definition of the Hamiltonian and the scheme may no longer apply. Since the Hamiltonian is quadratic in both the field observables and the ladder operators, we can again make the ansatz that the electromagnetic field operators are linear superpositions of creation and annihilation operators. Assuming that the Hamiltonian and field operators retain the same relationships with one another as their classical counterparts guarantees the validity of this linear superposition, since there must exist a linear transformation between any two sets of variables if a quantity (the Hamiltonian) can be independently written as a quadratic function of each set. Our linear superposition of creation and annihilation operators takes as coefficients the negative and positive frequency modes respectively with respect to the future-directed timelike killing vector $K^{\mu }$. The only modification we propose is the addition of a sum over spacetime sectors as introduced in the previous section. Including such flexibility will be essential in Section 4 when we quantise the electromagnetic field in an accelerated frame.

Thus the ansatz for the field operators becomes
(38)$$\begin{array}{ccc} \hat{E} & = & \sum_{\lambda =1,2}\int d^{3}k\left(\sum_{i}p_{k\lambda }^{\left(i\right)}{\hat{b}}_{k\lambda }^{\left(i\right)}+H.c.\right){\hat{e}}_{\lambda }\;,\\ \hat{B} & = & \sum_{\lambda =1,2}\int d^{3}k\left(\sum_{i}q_{k\lambda }^{\left(i\right)}{\hat{b}}_{k\lambda }^{\left(i\right)}+H.c.\right)\left(\hat{k}\times {\hat{e}}_{\lambda }\right)\;, \end{array}$$
where $p_{k\lambda }$ and $q_{k\lambda }$ are unknown positive frequency mode functions of all the spacetime coordinates, and ${\hat{e}}_{\lambda }$ is a unit polarisation vector orthogonal to the direction of the wave’s propagation at a point *x* in the spacetime. To determine the unknown mode functions, we demand that the expectation values of the operators satisfy the form of Maxwell’s equations explicit in E and B that derives from
(39)$$\frac{1}{\sqrt{\left|g\right|}}\partial_{\mu }\left(\sqrt{\left|g\right|}F^{\mu \nu }\right)=0\;\mathrm{and}\;\partial_{[\sigma }F_{\mu \nu ]}=0\;.$$
In general, this could be highly nontrivial and is indeed the greatest obstacle to a simple implementation of the scheme. Solving wave equations on curved spacetimes is a difficult task [49], so we would like to again follow the Minkowskian scheme and simplify the task by using a Heisenberg equation of motion.

To get around the manifest noncovariance of Equation (Equation 13), we note that since $\hat{H}$ generates a unitary group that implements time translation symmetry on the Fock space, the equation is a geometric expression of the fact that time evolution of operators is generated by the system’s Hamiltonian [52]. Considering the effect of an infinitesimal Poincaré transformation on an observable, $\hat{O}$ thus gives
(40)$$\partial_{\mu }\hat{O}=-i\left[\hat{O},{\hat{P}}_{\mu }\right]\;,$$
from which Equation (Equation 13) can be obtained as the 0th component [39,71,72]. Generalising this expression to curved spacetimes is then a simple matter of applying the minimal-coupling principle, giving
(41)$$\nabla_{\mu }\hat{O}=-i\left[\hat{O},{\hat{P}}_{\mu }\right]\;.$$
However, it is common to only consider evolution due to the Hamiltonian, in which case the Heisenberg equation is made covariant by using a proper time derivative to give [73,74,75]
(42)$$\frac{d\hat{O}}{d\tau }=-i\left[\hat{O},\hat{H}\right]\;.$$
Both approaches are used in the literature as covariant generalisations of the Heisenberg equation, yet they do not immediately appear to give the same results. To connect the two, we multiply Equation (Equation 41) by a tangent vector,
(43)$$U^{\mu }\nabla_{\mu }\hat{O}=-i\left[\hat{O}U^{\mu }{\hat{P}}_{\mu }-U^{\mu }{\hat{P}}_{\mu }\hat{O}\right]\;,$$
where we have assumed that it commutes with all the operators. Along a curve $x^{\alpha }$ the directional derivative of any given tensor T is $\frac{dT}{d\lambda }=\frac{dx^{\alpha }}{d\lambda }\nabla_{\alpha }T=U^{\alpha }\nabla_{\alpha }T\;,$ where λ is any affine parameter. The case λ=τ promotes $U^{\mu }$ to the four velocity. For a particle on a stationary spacetime, in its rest frame $U_{\mu }P^{\mu }=H$, and as this is a scalar this holds in any frame. Thus one obtains Equation (Equation 42), which is the proper time covariant Heisenberg equation of motion.

Our generalised quantisation scheme will apply this covariant Heisenberg equation to the expectation value $\langle \hat{O}\rangle$ of a general state in the photon Fock space |ψ〉,
(44)$$\nabla_{0}\langle \hat{O}\rangle =-i\langle \left[\hat{O},\hat{H}\right]\rangle \;.$$
This gives the temporal evolution in the wave equations resulting from Equation (Equation 39), where $\hat{H}$ is taken as the field Hamiltonian of Equation (Equation 37). If the form of Maxwell’s equations on the spacetime can be obtained and solved for the expectation values of the field operators using this procedure, the constant terms are determined by demanding that
(45)$$\hat{H}\equiv \frac{1}{2}\int_{\Sigma }d^{3}x\left({\hat{E}}^{2}+{\hat{B}}^{2}\right)\sqrt{\gamma }n_{0}K^{0}$$
on the spacelike hypersurface Σ. As the integration over this hypersurface is independent of the choice of surface and is constant, this holds for all time. In this manner, the unknown modes in Equation (Equation 38) can be determined and the electromagnetic field on a static, 4-dimensional Lorentzian manifold can be quantised.

#### 3.3.4. Summary of Scheme

Let us reflect on our construction. We have taken the Minkowskian gauge-independent electromagnetic field quantisation scheme in Section 2.3 and lifted it onto a static Lorentzian manifold with metric $g_{\mu \nu }$. Assuming the existence of detectable photons, the presence of a global timelike Killing vector allowed the definition of positive and negative frequency modes and thus the introduction of a well-defined particle Fock space, with general photon states labelled by their physical polarisation λ and wave-vector k. We introduced a ladder-operator structure for the Fock space, and using the approximation of Friis et al. [16] argued that this Fock space is associated with the field Hamiltonian of Equation (Equation 37) for minimal coupling to the background geometry.

The fact that both the field Hamiltonian $\hat{H}$ and the classical Hamiltonian *H* of Equation (Equation 31) were quadratic in the ladder operators or field strengths respectively allowed the proposal of a linear ansatz for the electric and magnetic field operators in terms of unknown wave modes. The scheme is then restricted to the specific manifold in question by demanding that the expectation values of these operators satisfy the modified Maxwell equations deriving from Equation (Equation 39), which introduces an explicit metric dependence to the scheme. To facilitate solving the potentially nontrivial Maxwell equations we use a form of the covariant Heisenberg equation, which we expect from work in Minkowski space to then uniquely determine the functional form of the modes in the operator ansatz. To determine all constants in these modes we demand that if we promote the classical Hamiltonian to an operator, upon substitution of the field operators the field Hamiltonian is regained.

By building off an already explicitly gauge-independent scheme, our method has the advantage of offering a gauge-independent and covariant route to the derivation of the Hamiltonian $\hat{H}$ and the electric and magnetic field observables, $\hat{E}$ and $\hat{B}$, respectively, on curved spacetimes. However, so far the only justification we have that this field quantisation scheme will give a physical result is based on its progenitor in Minkowski space. To test the consistency of our approach with other field quantisation schemes, we now consider a specific non-Minkowskian spacetime as an example and show that standard physical results are reproduced.

## 4. Electromagnetic Field Quantisation in an Accelerated Frame

In this section we apply the general formalism developed above to a specific example: 1-dimensional acceleration in Minkowski space. This situation is interesting as the noninertial nature of this motion leads to observers having different notions of particle states, and is thus often considered first in developments of quantum field theory in curved spacetime. It is also the situation most easily accessible to experimental tests. We must note that Soldati and Specchia [34] have emphasised photon propagation in accelerated frames remains conceptually nontrivial due to the separation of physical and nonphysical polarisation modes arising from standard quantisation techniques. Here we avoid these issues by only considering motion in the direction of acceleration (1D propagation) [33,34], and also by avoiding the use of canonical quantisation and immediately considering the physical degrees of freedom.

### 4.1. Rindler Space

An observer in Minkowski space M accelerating along a one-dimensional line with proper acceleration α appears to an inertial observer to travel along a hyperbolic worldline
(46)$$x^{\mu }=\left(\frac{1}{\alpha }\sinh\left(\alpha \tau \right),\frac{1}{\alpha }\cosh\left(\alpha \tau \right),0,0\right),\;x^{\mu }x_{\mu }=-\frac{1}{\alpha^{2}}\;,$$
where τ is the accelerating observer’s proper time. As the proper acceleration α→∞, the hyperbolic worldline of Equation (Equation 46) becomes asymptotic to the null lines of M, x=t for t>0 and x=−t for t<0. The interior region in which the hyperbola resides is defined by |t|<x and is called the Right Rindler wedge (RR); if |t|<−x we have the Left Rindler wedge (LR). The union of both wedges yields the Rindler space R, which is a static globally hyperbolic spacetime [58].

More concretely, we can obtain Rindler space by the coordinate transformation
(47)$$\begin{array}{c} t=\pm \rho \sinh(\alpha \zeta )\;,\;x=\pm \rho \cosh(\alpha \zeta )\;,\;y=y\;,\;z=z\;, \end{array}$$
where we call the coordinates (ζ,ρ,y,z) polar Rindler coordinates, with positive signs labelling points in RR and negative signs labelling those in LR [76]. In this coordinate system, the metric associated with the frame of accelerating observer $O^{\prime }$ is [34,56,62]
(48)$$ds^{2}=\alpha^{2}\rho^{2}d\zeta^{2}-d\rho^{2}-dy^{2}-dz^{2}\;.$$
The right Rindler wedge is covered by the set of all uniformly accelerated motions such that $\alpha^{-1}\in R^{+}$, and the boundaries of Rindler space are Cauchy horizons for the motion of $O^{\prime }$ [61,63].

Many studies of this spacetime choose to introduce conformal Rindler coordinates (ξ,η,y,z) [58,76], defined by the coordinate transformation
(49)$$t=\pm a^{-1}e^{a\xi }\sinh\left(a\eta \right)\;,\;x=\pm a^{-1}e^{a\xi }\cosh\left(a\eta \right)\;,\;y=y\;,\;z=z\;,$$
where a∈R is a positive constant such that $ae^{-a\xi }=\alpha$, so the proper time τ of $O^{\prime }$ relates to η as $\tau =e^{a\xi }\eta$. The two coordinate systems for R hence relate as
(50)$$\rho =a^{-1}e^{a\xi }\;\mathrm{and}\;\alpha \zeta =a\eta \;.$$
Lines of constant Rindler coordinates are shown in Figure 1. Rindler space can thus also be associated with the metric line element
(51)$$ds^{2}=e^{2a\xi }\left(d\eta^{2}-d\xi^{2}\right)-dy^{2}-dz^{2}\;.$$


These coordinates are useful because worldlines with ξ=0 have constant acceleration a=α [58].

From the discussion of Killing vectors in Section 3.3, it is immediate that since the metric components are independent of ζ or η in the respective coordinate systems, $\partial_{\eta }\equiv \frac{\alpha }{a}\partial_{\zeta }$ is a Killing field for R, and moreover the field is timelike. However in LR the field is orientated in the past time direction, so the future-directed timelike killing vector in this wedge is $\partial_{\left(-\eta \right)}=-\partial_{\eta }\equiv -\frac{\alpha }{a}\partial_{\zeta }$. To deal with this when considering wave propagation, one must introduce two disjoint sets of positive frequency modes $f_{k}^{\left(i\right)}$, i=L,R. These satisfy
(52)$$\begin{array}{c} \partial_{\eta }f_{k}^{\left(R\right)}=-i\omega_{k}f_{k}^{\left(R\right)}\;\;\mathrm{and}\;\;-\partial_{\eta }f_{k}^{\left(L\right)}=-i\omega_{k}f_{k}^{\left(L\right)}\;, \end{array}$$
so each set is positive frequency with respect to its appropriate future-directed timelike Killing vector. These sets and their conjugates form a complete basis for solutions of the wave equation on R [48,51].

As a region of Minkowski space Rindler space is a flat spacetime with no matter content [64]. Despite this, because of the spacetime’s noninertial nature covariant considerations must be applied when working in R. For example the naïve divergence $\partial_{\mu }A^{\mu }\neq \partial^{\mu }A_{\mu }$ as required by Lorentz invariance, and we have non-zero Christoffel symbols
(53)$$\Gamma_{\xi \xi }^{\xi }=\Gamma_{\eta \eta }^{\xi }=\Gamma_{\eta \xi }^{\eta }=\Gamma_{\xi \eta }^{\eta }=a\;.$$
With the Christoffel symbols covariant derivatives $\nabla_{\mu }$ can be taken, and the timelike Killing vector fields $\partial_{\eta }$ and $\partial_{\left(-\eta \right)}$ can be shown to formally satisfy Equation (Equation 20).

### 4.2. Electromagnetism in Rindler Space

To apply our covariant gauge-independent quantisation scheme to accelerating frames, we need to consider classical electromagnetism in Rindler space. Our starting point, the field strength tensor, takes the standard form
(54)$$F_{\mu \nu }^{R}=\left(\begin{array}{cccc} 0 & E_{R}^{1} & E_{R}^{2} & E_{R}^{3}\\ -E_{R}^{1} & 0 & -B_{R}^{3} & B_{R}^{2}\\ -E_{R}^{2} & B_{R}^{3} & 0 & -B_{R}^{1}\\ -E_{R}^{3} & -B_{R}^{2} & B_{R}^{1} & 0 \end{array}\right)\;.$$
The explicit relations between the Rindler fields and those in Minkowski space are given in Appendix A. These relations are taken to define the fields in the accelerated frame. For the Maxwell equation we need the contravariant field strength tensor $F^{\mu \nu }=g^{\mu \sigma }g^{\nu \rho }F_{\sigma \rho }$. Because of the metric contractions this is explicitly coordinate dependent. In conformal coordinates we have
(55)$$F_{R}^{\mu \nu }=\left(\begin{array}{cccc} 0 & -E_{R}^{1}e^{-4a\xi } & -E_{R}^{2}e^{-2a\xi } & -E_{R}^{3}e^{-2a\xi }\\ E_{R}^{1}e^{-4a\xi } & 0 & -B_{R}^{3}e^{-2a\xi } & B_{R}^{2}e^{-2a\xi }\\ E_{R}^{2}e^{-2a\xi } & B_{R}^{3}e^{-2a\xi } & 0 & -B_{R}^{1}\\ E_{R}^{3}e^{-2a\xi } & -B_{R}^{2}e^{-2a\xi } & B_{R}^{1} & 0 \end{array}\right).$$
The polar coordinate form of this equation can be found in Appendix A.

The Maxwell equations that incorporate the spacetime’s nontrivial geometry now follow from Equation (Equation 39). In Rindler space and conformal coordinates, $g=-e^{4a\xi }$. Thus we obtain
(56)$$\begin{array}{c} e^{-2a\xi }\partial_{\xi }E_{R}^{1}-2aE_{R}^{1}e^{-2a\xi }+\partial_{y}E_{R}^{2}+\partial_{z}E_{R}^{3}=0\;,\\ e^{-2a\xi }\partial_{\eta }E_{R}^{1}=\partial_{y}B_{R}^{3}-\partial_{z}B_{R}^{2}\;,\\ \partial_{\eta }E_{R}^{2}=e^{2a\xi }\partial_{z}B_{R}^{1}-\partial_{\xi }B_{R}^{3}\;,\\ \partial_{\eta }E_{R}^{3}=\partial_{\xi }B_{R}^{2}-e^{2a\xi }\partial_{y}B_{R}^{1}\;. \end{array}$$
The set of equations deriving from the Bianchi identity are exactly the same as in flat space; these are listed in Appendix A, along with the full Maxwell equations in polar coordinates.

### 4.3. Field Quantisation in Rindler Space

Knowing how classical electric and magnetic amplitudes evolve in Rindler space, we are now in a position to derive the Hamiltonian $\hat{H}$ and the electric and magnetic field observables, $\hat{E}$ and $\hat{B}$, respectively, of the quantised electromagnetic field in Rindler space R. For simplicity, we are only interested in photons which propagate along one spatial dimension. Suppose they travel along the ξ axis in conformal or along the ρ axis in polar coordinates, which from Equation (Equation 50) are proportional and thus equivalent. Thus photon modes will have a wave-number *k* and a polarisation λ=1,2 as their labels. Working in only one dimension, we have avoided the necessity to introduce more complicated polarisations [34].

Unfortunately, the general states in Equation (Equation 9) are complicated in R by the existence of different future-directed timelike killing vectors in the two Rindler wedges, with $\partial_{\eta }$ in RR and $-\partial_{\eta }$ in LR. Hence there need to be two sets of positive frequency modes for solutions of the wave equation on the spacetime. There will thus be two distinct Fock spaces representing the particle content in LR and RR. A general particle number state for light propagating in one dimension in R will hence be
(57)$$\underset{\lambda =1,2}{⨂}\overset{\infty }{\underset{k=-\infty }{⨂}}|n_{k\lambda }^{L},n_{k\lambda }^{R}\rangle \;,$$
with $n_{k\lambda }^{L}$ being the number of photons in LR and $n_{k\lambda }^{R}$ being the number of photons in RR. Thus the physical energy eigenstates are in general degenerate and the Hamiltonian must satisfy
(58)$$\hat{H}\;|n_{k\lambda }^{L},n_{k\lambda }^{R}\rangle =\left[\omega_{k}\left(n_{k\lambda }^{L}+n_{k\lambda }^{R}\right)+H_{0}\right]|n_{k\lambda }^{L},n_{k\lambda }^{R}\rangle ,$$
with integer values for both $n_{k\lambda }^{L}$ and $n_{k\lambda }^{R}$. This suggests that the field Hamiltonian $\hat{H}$ of Equation (Equation 37) has to be expressed in terms of independent ladder operators for both wedges. Hence, it can be written as
(59)$$\hat{H}=\sum_{\lambda =1,2}\int_{-\infty }^{\infty }dk\left[\omega_{k}\left({\hat{b}}_{k\lambda }^{R†}\;{\hat{b}}_{k\lambda }^{R}-{\hat{b}}_{k\lambda }^{L†}\;{\hat{b}}_{k\lambda }^{L}\right)+H_{0}\right]\;,$$
where the $E_{\omega_{k}}$ factor of Equations (Equation 34) and (Equation 52) give the relative sign between the left and right sectors. As we are considering photons propagating along ξ or ρ, and photons are electromagnetic waves, the electric and magnetic fields must be in the transverse spatial dimensions y,z that are unaffected by the acceleration and thus identical to their Minkowski counterparts. As described in Section 2.3, the polarisation basis states correspond to choices of these fields. Here we choose
(60)$$E,B=\left\{\begin{array}{cc} \left(0,E,0),\;(0,0,B\right) & \;\lambda =1\\ \left(0,0,E),\;(0,-B,0\right) & \;\lambda =2\;, \end{array}\right.$$
where *E* and *B* are scalar functions of (ζ,ρ) or (η,ξ). With this choice of fields, the Rindler–Maxwell equations of Equation (Equation 56) reduce to
(61)$$\partial_{\eta }E=-\partial_{\xi }B\;,\;\partial_{\xi }E=-\partial_{\eta }B\;,$$
for conformal Rindler coordinates, and from Equation (Equation 89) to
(62)$$\frac{1}{\rho^{2}\alpha^{2}}\partial_{\zeta }E=-\left(\partial_{\rho }B+\frac{1}{\rho }B\right),\;\partial_{\rho }E=-\partial_{\zeta }B\;,$$
for polar coordinates. Both sets of equations hold in both LR and RR. The conformal expressions are now identical to the 1D Minkowski propagation considered in [35]. It should be emphasised that the apparent simplicity is a result of demanding 1-dimensional propagation along the accelerated spatial axis and choosing convenient polarisations.

The noninertial nature of Rindler space still requires care; recall from Equation (Equation 31) that to determine the classical electromagnetic Hamiltonian, we require a timelike Killing vector field. We must also choose a spacelike hypersurface Σ with normal vector $n^{\mu }$ and induced metric $\gamma_{ij}$ to integrate over. In conformal Rindler coordinates, we know that the timelike Killing vector field is $K=\partial_{\eta }$, so $K^{\mu }=\delta_{\eta }^{\mu }$. Choosing Σ as being the hypersurface defined by η=0 allows us to continue using the spatial coordinates $x^{i}=\left(\xi ,y,z\right)$. Hence, the full conformal Rindler metric of Equation (Equation 51) implies $\gamma =\det\left(\gamma_{ij}\right)=e^{-2a\xi }$. Finally, since Σ is spacelike, $n^{\mu }$ is normalised to +1, so
(63)$$1=g_{\mu \nu }n^{\mu }n^{\nu }=e^{2a\xi }{\left(n^{0}\right)}^{2}\;,$$
giving $n^{0}=e^{-a\xi }$ [48]. Hence the Hamiltonian in Rindler space is
(64)$$\begin{array}{cc} H & =\frac{1}{2}\int d\xi \left(E^{2}+B^{2}\right)e^{a\xi }\;e^{-a\xi }\;\delta_{\eta }^{\eta }\\ & =\frac{1}{2}\int d\xi \left(E^{2}+B^{2}\right), \end{array}$$
so the initial apparent simplicity holds.

Following our general prescription, we again make the ansatz that the field operators are linear superpositions of the relevant ladder operators. As we are considering 1-dimensional propagation with the electric and magnetic field vectors E and B, respectively, as specified in Equation (Equation 60), we need only apply the ansatz to the scalar components *E* and *B* for quantisation, giving
(65)$$\begin{array}{c} \hat{E}=\sum_{\lambda =1,2}\int_{-\infty }^{\infty }dk\left(p_{k\lambda }^{L}{\hat{b}}_{k\lambda }^{L}+p_{k\lambda }^{R}{\hat{b}}_{k\lambda }^{R}+H.c.\right),\\ \hat{B}=\sum_{\lambda =1,2}\int_{-\infty }^{\infty }dk\left(q_{k\lambda }^{L}{\hat{b}}_{k\lambda }^{L}+q_{k\lambda }^{R}{\hat{b}}_{k\lambda }^{R}+H.c.\right), \end{array}$$
where $p_{k\lambda }^{i}$ and $q_{k\lambda }^{i}$ are unknown functions of (η,ξ), and i=L,R for LR and RR respectively. Since the left and the right wedges of R are causally disjoint, we can demand that modes in different wedges are orthogonal with respect to the inner product in Equation (Equation 36) [48]. Explicitly this yields
(66)$$\begin{array}{c} \langle p_{k\lambda }^{L},p_{k^{\prime }\lambda^{\prime }}^{R}\rangle =\int_{-\infty }^{\infty }dk\;p_{k\lambda }^{*L}p_{k^{\prime }\lambda^{\prime }}^{R}=0\;,\\ \langle p_{k\lambda }^{*L},p_{k^{\prime }\lambda^{\prime }}^{R}\rangle =\int_{-\infty }^{\infty }dk\;p_{k\lambda }^{L}p_{k^{\prime }\lambda^{\prime }}^{R}=0 \end{array}$$
with similar expressions for $q_{k\lambda }^{i}$. To determine all the modes, we follow the recipe of Section 3.3 and demand that the expectation values of these field operators satisfy Equations (Equation 61) and (Equation 62).

From now on we will work in the conformal Rindler coordinates (η,ξ) due to the wonderful simplicity of their Maxwell equations. One could of course also use polar coordinates, and indeed one can show that this yields the same results in this set for the case a=α. To determine temporal evolution we use the Heisenberg equation, which, as the time coordinate is η in this system and our observables $\hat{O}$ are scalars, is
(67)$$\partial_{\eta }\hat{O}=-i\left[\hat{O},\hat{H}\right].$$
Following our prescription, we compare expectation values of the ladder operators for spatial derivatives and time evolution from Heisenberg’s equation by using our form of Maxwell’s equations. In this case, using Equations (Equation 61) this procedure gives the relations
(68)$$\begin{array}{ccc} \partial_{\xi }q_{k\lambda }^{i} & = & i\omega_{k}p_{k\lambda }^{i}\;, \end{array}$$
(69)$$\begin{array}{ccc} \partial_{\xi }p_{k\lambda }^{i} & = & i\omega_{k}q_{k\lambda }^{i}\;. \end{array}$$
Solving for $p_{k\lambda }^{i}$ we of course just obtain the wave equation, $\left(\partial_{\xi }^{2}+k^{2}\right)p_{k\lambda }^{i}=0$, when we consider free, on-shell photons with $k^{2}=\omega_{k}^{2}$. This equation admits separable solutions $p_{k\lambda }^{i}=\chi_{k\lambda }^{i}\left(\eta \right)P_{k\lambda }^{i}\left(\xi \right)$, so as there are no temporal derivatives we lose all temporal information. Writing the spatial solution is trivial:
(70)$$P_{k\lambda }^{i}=J_{\lambda }^{i}e^{ik\xi }+K_{\lambda }^{i}e^{-ik\xi }\;,$$
where $J_{\lambda }^{i},K_{\lambda }^{i}\in C$. To determine the temporal dependence of $\chi_{k\lambda }\left(\eta \right)$ we use that positive frequency Rindler modes must satisfy Equation (Equation 52). The two modes $p_{k\lambda }^{L}$ and $p_{k\lambda }^{R}$ must both be positive frequency with respect to the future-direction of $\partial_{\eta }$ as they are coefficients of annihilation operators [40]. Thus the difference between them will be in their time dependence. This gives that we must have
(71)$$\chi_{k\lambda }^{L}=e^{i\omega_{k}\eta },\;\chi_{k\lambda }^{R}=e^{-i\omega_{k}\eta }.$$
This difference is a direct result of the two Rindler wedges having different future-directed timelike Killing vectors. Thus, in all, we have
(72)$$\begin{array}{cc} p_{k\lambda }^{R}\left(\eta ,\xi \right) & =U_{\lambda }^{R}e^{i(k\xi -\omega_{k}\eta )}+V_{\lambda }^{R}e^{-i(k\xi +\omega_{k}\zeta )}\;,\\ p_{k\lambda }^{L}\left(\eta ,\xi \right) & =U_{\lambda }^{L}e^{i(k\xi +\omega_{k}\eta )}+V_{\lambda }^{L}e^{-i(k\xi -\omega_{k}\eta )}\;. \end{array}$$
We can then easily obtain the $q_{k\lambda }^{i}$ solutions from Equation (Equation 68) as
(73)$$\begin{array}{cc} q_{k\lambda }^{R}\left(\eta ,\xi \right) & =\frac{k}{\omega_{k}}\left[U_{\lambda }^{R}e^{i(k\xi -\omega_{k}\eta )}-V_{\lambda }^{R}e^{-i(k\xi +\omega_{k}\eta )}\right],\\ q_{k\lambda }^{L}\left(\eta ,\xi \right) & =\frac{k}{\omega_{k}}\left[U_{\lambda }^{L}e^{i(k\xi +\omega_{k}\eta )}-V_{\lambda }^{L}e^{-i(k\xi -\omega_{k}\eta )}\right].\;\; \end{array}$$
We now seek to determine the unknown coefficients in these expressions. Similarly to Section 2.3, first note that wave modes propagating in the positive ξ direction in R should be functions of $\left(k\xi -\omega_{k}\eta \right)$ in RR where $\partial_{\eta }$ is the future-directed timelike Killing vector, and functions of $\left(k\xi +\omega_{k}\eta \right)$ in LR where it is $-\partial_{\eta }$. Similarly, modes propagating in the negative ξ direction should be functions of $\left(k\xi +\omega_{k}\eta \right)$ in RR and functions of $\left(k\xi -\omega_{k}\eta \right)$ in LR. These conditions imply $V^{R}=V^{L}=0$.

We then determine the remaining constants by demanding that the classical and the quantised field Hamiltonians are equivalent, as in Equation (Equation 45). Since $\hat{H}$ is quadratic in the electric and magnetic field operators, we obtain cross terms between LR and RR modes during the calculation. Integrating over such terms gives the inner products in Equation (Equation 66), but as modes in the different wedges are orthogonal these terms are identically 0, so there are no physical cross terms. Then after some algebra and relying on the integral definition of the delta function, we arrive at
(74)$$\begin{array}{c} \hat{H}=2\pi \sum_{\lambda =1,2}\int_{-\infty }^{\infty }dk\left[|U_{\lambda }^{R}{|}^{2}\left(2{\hat{b}}_{k\lambda }^{†R}\;{\hat{b}}_{k\lambda }^{R}+\delta \left(0\right)\right)+{\left|U_{\lambda }^{L}\right|}^{2}\left(2{\hat{b}}_{k\lambda }^{†L}\;{\hat{b}}_{k\lambda }^{L}+\delta \left(0\right)\right)\right]\;, \end{array}$$
where we have used the commutation relations in Equation (Equation 25). As in Section 2.3, to finally determine the constant terms and zero-point energy we compare with Equation (Equation 59) which yields
(75)$$|U_{\lambda }^{R}{|}^{2}=\frac{\omega_{k}}{4\pi }\;,\;{\left|U_{\lambda }^{L}\right|}^{2}=\frac{\omega_{k}}{4\pi }\;,\;H_{0}=\int_{-\infty }^{\infty }dk\;\omega_{k}\;\delta \left(0\right)\;.$$
To obtain our final expressions for the electric and magnetic field operators we arbitrarily choose the phases of both $U_{\lambda }^{R}$ and $U_{\lambda }^{L}$ to give consistency with standard Minkowskian results, and multiply the electric field operator by polarisation unit vector ${\hat{e}}_{\lambda }$ and the magnetic field operator by $\hat{k}\times {\hat{e}}_{\lambda }$. Thus, in all, we obtain the final results
(76)$$\begin{array}{ccc} \hat{E} & = & \;i\sum_{\lambda =1,2}\int_{-\infty }^{\infty }dk\;\sqrt{\frac{\omega_{k}}{4\pi }}\;\left[e^{i(k\xi -\omega_{k}\eta )}\;{\hat{b}}_{k\lambda }^{R}+e^{i(k\xi +\omega_{k}\eta )}\;{\hat{b}}_{k\lambda }^{L}+H.c.\right]\;{\hat{e}}_{\lambda }\;,\\ \hat{B} & = & -i\sum_{\lambda =1,2}\int_{-\infty }^{\infty }dk\;\sqrt{\frac{\omega_{k}}{4\pi }}\;\left[e^{i(k\xi -\omega_{k}\eta )}\;{\hat{b}}_{k\lambda }^{R}+e^{i(k\xi +\omega_{k}\eta )}\;{\hat{b}}_{k\lambda }^{L}+H.c.\right]\left(\hat{k}\times {\hat{e}}_{\lambda }\right)\;,\\ \hat{H} & = & \sum_{\lambda =1,2}\int_{-\infty }^{\infty }dk\;\omega_{k}\;\left[{\hat{b}}_{k\lambda }^{†R}{\hat{b}}_{k\lambda }^{R}+{\hat{b}}_{k\lambda }^{†L}{\hat{b}}_{k\lambda }^{L}+\delta \left(0\right)\right]\;. \end{array}$$
These three operators are very similar to the electric and magnetic field operators $\hat{E}$ and $\hat{B}$, respectively, and $\hat{H}$ in Equations (Equation 11) and (Equation 14) in Minkowski space. When moving in only one dimension, the orientation of the electric and magnetic field amplitudes is still pairwise orthogonal and orthogonal to the direction of propagation. However, the electromagnetic field has become degenerate and additional degrees of freedom which correspond to different Rindler wedges have to be taken into account in addition to the wave numbers *k* and the polarisations λ of the photons. Finally, instead of depending on kx, the electric and magnetic field observables now depend on $k\xi \pm \omega_{k}\eta$, i.e., they depend not only on the position but also on the amount of time the observer has been accelerating in space and on their acceleration. Most importantly, Equation (76) can now be used as the starting point for further investigations into the quantum optics of an accelerating observer [5,36,46], and is expected to find immediate applications in relativistic quantum information [13,14,15,16,17,18,19,20,21,69].

### 4.4. The Unruh Effect

As an example and to obtain a consistency check, we now verify that our results give the well-established Unruh effect [55,56,58,59]. This effect predicts that an observer with uniform acceleration α in Minkowski space measures the Minkowski vacuum as being a pure thermal state with temperature
(77)$$T_{\mathrm{Unruh}}=\frac{\alpha }{2\pi }\;.$$
Deriving this result relies on being able to switch between modes in Minkowski and modes in Rindler space, which requires a Bogolubov transformation. This transformation allows us to switch between the modes of different coordinate frames and generally transforms a vacuum state to a thermal state [57,77]. For a field expansion in two complete sets of basis modes, $\phi =\sum_{i}{\hat{a}}_{i}f_{i}+{\hat{a}}_{i}^{†}f_{i}^{*}=\sum_{j}{\hat{b}}_{j}g_{j}+{\hat{b}}_{j}^{†}g_{j}$, this relates the modes as
(78)$$\begin{array}{c} g_{i}=\sum_{j}\alpha_{ij}f_{j}+\beta_{ij}f_{j}^{*}\;,\\ f_{i}=\sum_{j}\alpha_{ji}^{*}g_{j}-\beta_{ji}g_{j}^{*}\;, \end{array}$$
where $\alpha_{ij}$ and $\beta_{ij}$ are the Bogolubov coefficients [58]. Knowing these coefficients also allows the associated particle Fock spaces to be related,
(79)$$\begin{array}{c} {\hat{a}}_{i}=\sum_{j}\alpha_{ji}{\hat{b}}_{j}+\beta_{ji}^{*}{\hat{b}}_{j}^{†}\;,\\ {\hat{b}}_{i}=\sum_{j}\alpha_{ij}^{*}{\hat{a}}_{j}-\beta_{ij}^{*}{\hat{a}}_{j}^{†}\;. \end{array}$$
For transforming between the Rindler and Minkowski Fock spaces, the coefficients can be calculated using coordinate relations in a method first introduced by Unruh [55].

Here our field modes are the expansions of the electric field operators in R and M with the Minkowski results taking the same functional form. Following the standard approach [48,51], our expressions for the field operators yield
(80)$$\begin{array}{c} \alpha_{LL}=\alpha_{RR}=\frac{1}{\omega_{k}}\;\sqrt{\frac{1}{2\sinh(\frac{\pi \omega_{k}}{a})}}\;e^{\frac{\pi \omega_{k}}{2a}}\;\\ \beta_{LR}=\beta_{RL}=\frac{1}{\omega_{k}}\;\sqrt{\frac{1}{2\sinh(\frac{\pi \omega_{k}}{a})}}\;e^{-\frac{\pi \omega_{k}}{2a}}\;. \end{array}$$
These immediately give the following relationship between the ladder operators.
(81)$$\begin{array}{cc} {\hat{b}}_{k\lambda }^{R} & =\frac{1}{\omega_{k}}\sqrt{\frac{1}{2\sinh(\frac{\pi \omega_{k}}{a})}}\left(e^{\frac{\pi \omega_{k}}{2a}}\;{\hat{c}}_{k\lambda }^{R}+e^{-\frac{\pi \omega_{k}}{2a}}\;{\hat{c}}_{-k\lambda }^{L†}\;\right)\;,\\ {\hat{b}}_{k\lambda }^{L} & =\frac{1}{\omega_{k}}\sqrt{\frac{1}{2\sinh(\frac{\pi \omega_{k}}{a})}}\left(e^{\frac{\pi \omega_{k}}{2a}}\;{\hat{c}}_{k\lambda }^{L}+e^{-\frac{\pi \omega_{k}}{2a}}\;{\hat{c}}_{-k\lambda }^{R†}\;\right). \end{array}$$
The ${\hat{c}}_{k\lambda }^{i}$ operators are associated with modes that can be purely expressed in terms of positive frequency Minkowski modes (from the form of the field operators in Cartesian coordinates). They must thus share the Minkowski vacuum, so ${\hat{c}}_{k}^{R}\;|0_{M}\rangle ={\hat{c}}_{k}^{L}\;|0_{M}\rangle =0$. Because we possess the Bogolubov transformation between Minkowski and Rindler space, we can now evaluate particle states seen by an observer in R, given by ${\hat{b}}_{k}^{i}$, in terms of a Minkowski Fock space given by $c_{k}^{i}$. In particular, evaluating the RR number operator on the Minkowski vacuum gives
(82)$$\begin{array}{c} \langle 0_{M}|{\hat{b}}_{k}^{R†}{\hat{b}}_{k}^{R}|0_{M}\rangle =\frac{1}{\omega_{k}^{2}}\;\frac{\delta (0)}{\exp(\frac{2\pi \omega_{k}}{a})-1}\;. \end{array}$$
This energy expectation value is the same as the energy expectation value of a thermal Planckian state with temperature $\frac{a}{2\pi }$. For the case a=α this is the prediction that exactly constitutes the Unruh effect, and thus verifies that the results of our quantisation scheme match known theoretical predictions. Having a≠α just corresponds to a redshift [48]. The external factor $1/\omega_{k}^{2}$ is different to that for a standard scalar field; this is just a remnant of the different normalisation of our electric field operator and does not affect the physical prediction, with such factors indeed sometimes appearing in the literature [49].

## 5. Conclusions

This paper generalises the physically-motivated quantisation scheme of the electromagnetic field in Minkowski space [35] to static spacetimes of otherwise arbitrary geometry. As shown in Section 3, such a generalisation requires only minimal modification of the original quantisation scheme in flat space. In order to assess the validity of the presented generalised approach, we apply our findings in Section 4 to the well understood case of Rindler space: the relevant geometry for a uniformly accelerating observer. Since this reproduces the anticipated Unruh effect, it supports the hypothesis that our approach is a consistent approach to the quantisation of the electromagnetic field on curved spacetimes.

The main strength of our quantisation scheme is its gauge-independence, i.e., its nonreliance on the gauge-dependent potentials of more traditional approaches. Instead it relies only on the experimentally verified existence of electromagnetic field quanta. As such, our scheme provides a more intuitive approach to field quantisation, while still relying on well established concepts and constructions in quantum field theory in curved space. Given this and the applicability of our results to accelerating frames in an otherwise flat spacetime, it seems likely that our approach can also be used to model more complex, but experimentally accessible, situations with applications, for example, in relativistic quantum information.

The specific case of Rindler space, as considered in this paper, led to equations with straightforward analytic solutions. This will likely not be true in more general settings, where the necessary wave equations will be nontrivial and will possibly require approximation or numerical solution. This fact is partially mitigated by our use of the Heisenberg equation, thereby reducing the necessary calculation to an ordinary differential equation and commutation relation, rather than a partial differential equation. Furthermore, recall that the scheme laid out in this paper is a generalisation of that in flat space to the case of static curved spacetimes. This simplified the definition and construction of the quantisation scheme, due to our reliance on spacelike hypersurfaces. When applied to the more general case of stationary spacetimes, the correct prescription of the scheme becomes less clear and will require further theoretical development.

## Figures and Tables

**Figure 1 entropy-21-00844-f001:**
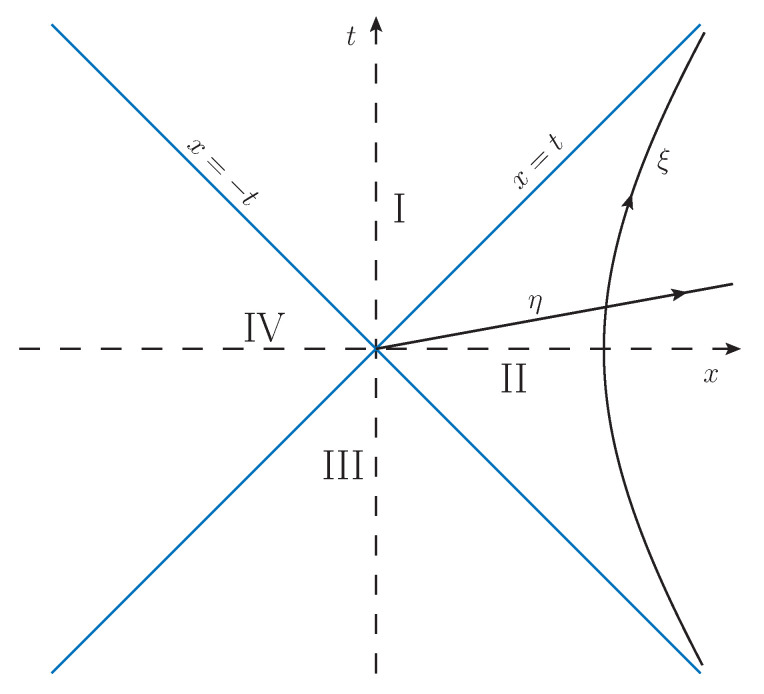
Depiction of a 2-dimensional Minkowski space M. Regions I and III are the future and past light cones of the observer O at the origin, while regions II and IV are the right Rindler wedge (RR) and left Rindler wedge (LR) respectively. The worldline of a uniformly accelerated observer with acceleration α is the displayed line of constant conformal Rindler coordinate ξ.

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
