# Peer review of "A Physically-Motivated Quantisation of the Electromagnetic Field on Curved Spacetimes"

_entropy, 2019, doi:10.3390/e21090844_

Round 1

Reviewer 1 Report

The paper verses on the quantisation of a Maxwell field in a curved space, specifically on a Rindler spacetime. The authors  that they obtained a unification of such field on Einstein strong gravity. On the other hands, some points below must be  considered: 

*What are the advantages of this framework?why dont the authors use the Yang-Mills field?

*How to compatibilize the group structure in such scheme, since the quantization is made in a Hibert space and the gravitation one (where the diffeomorphism group is defined)?Why not to begin with a definition of the gravitational field (a graviton) in the Hilbert space? 

*How should the renormalization problem be dealt in such scheme since the Einstein gravity is a non-renormalizable theory? (I suggest the auhtor check refs. A. O. Barvinsky, A. Yu. Kamenshchik, and I. P. Karmazin Phys. Rev. D 48, 3677 and K.S. Stelle, Phys. Rev. D 16, 953 (1977)).

*what are the gauge-Invariant observables?

Comparing the inital hypothesis and the results, the paper can be reconsidered to publication after the authors discuss the former raised points.

Author Response

Thank you to the reviewers for their useful comments on our manuscript. We have found the comments valuable when revising our paper. Please see below, how we have addressed each of the individual comments. 

Reviewer: What are the advantages of this framework? Why don’t the authors use the Yang-Mills field?

Reply: The aim of the paper is not to quantise the gravitational field but to quantise the electromagnetic field in a fixed background spacetime. This is now further clarified on lines 43-47, 66-69. Specifically, we do not consider the back-reaction of the field on the spacetime geometry nor a quantised gravitational field. Since the only quantisation considered is that of the Abelian electromagnetic field, the non-Abelian Yang-Mills field is not applicable. 

-

Reviewer: How to compatibilize the group structure in such scheme, since the quantization is made in a Hilbert space and the gravitation one (where the diffeomorphism group is defined)? Why not to begin with a definition of the gravitational field (a graviton) in the Hilbert space?

Reply: Since the paper considers only quantisation of the electromagnetic field and treats spacetime as fixed and classical, there is no graviton in our scheme. This is now further clarified on lines 43-47, 66-69. The only quantum objects in the scheme are the photons and so our Hilbert space need only contain photon states. 

-

Reviewer: How should the renormalization problem be dealt in such scheme since the Einstein gravity is a non-renormalizable theory?

Reply: Our approach is one of quantum field theory in curved space. We do not attempt to quantise the gravitational field and so the non-renormalisability of the gravitational field is not a concern. This is now further clarified on lines 43-47, 66-69.

-

Reviewer: What are the gauge-invariant observables?

Reply: This has been clarified in the final paragraph of section 2.3 (lines 147-150).

Reviewer 2 Report

The manuscript is well written, the content is good. I would suggest some minor changes in the presentation.

Line 146 when you write "general curved spacetimes with metric $\g_{\mu \nu}" why not simply "general curved spacetimes" instead?

Eq. 21 +ve and -ve please write instead positive and negative.

paragraph 3.3.3 

please give more evidence to your choice and explain better what are the consequences of your assumption
"we can again make the ansatz that the electromagnetic field operators are linear superpositions of creation and annihilation operators..." 

and briefly comment the resulting scenario if your ansatz could not be applied.

I would suggest to publish the present manuscript after these minor changes.

Author Response

Thank you to the reviewers for their useful comments on our manuscript. We have found the comments valuable when revising our paper. Please see below, how we have addressed each of the individual comments. 

Reviewer: Line 146 when you write “general curved spacetimes with metric $g_{\mu\nu}$” why not simply “general curved spacetimes” instead?

Reply: This change has been made.

-

Reviewer: Eq. 21 +ve and -ve please write instead positive and negative

Reply: This change has been made.

-

Reviewer: Paragraph 3.3.3: Please give more evidence to your choice and explain better what are the consequences of your assumption… And briefly comment on the resulting scenario if your ansatz could not be applied

Reply: Further justification for the ansatz is now given in this paragraph (lines 211-225).

Round 2

Reviewer 1 Report

The authors did the necessary efforts to improve the presentation and they answered all points raised in the previous report. Accordingly, I reccomend as acceptable this manuscript to publication.